# TreeCluster: Clustering biological sequences using phylogenetic trees

**Metin Balaban**[1], **Niema Moshiri**[1], **Uyen Mai**[2], **Xingfan Jia**[3], **Siavash Mirarab**[4]*

**1** Bioinformatics and Systems Biology Graduate Program, UC San Diego, La Jolla, CA 92093, United States of America, **2** Computer Science and Engineering, UC San Diego, La Jolla, CA 92093, United States of America, **3** Department of Mathematics, UC San Diego, La Jolla, CA 92093, United States of America, **4** Department of Electrical and Computer Engineering, UC San Diego, La Jolla, CA 92093, United States of America

* smirarab@ucsd.edu

**Data Availability Statement:** The datasets and scripts used are all available at: http://doi.org/10.5281/zenodo.3358386.

**Funding:** This work was supported by the National Institutes of Health (NIH) subaward

## Abstract

Clustering homologous sequences based on their similarity is a problem that appears in many bioinformatics applications. The fact that sequences cluster is ultimately the result of their phylogenetic relationships. Despite this observation and the natural ways in which a tree can define clusters, most applications of sequence clustering do not use a phylogenetic tree and instead operate on pairwise sequence distances. Due to advances in large-scale phylogenetic inference, we argue that tree-based clustering is under-utilized. We define a family of optimization problems that, given an arbitrary tree, return the minimum number of clusters such that all clusters adhere to constraints on their heterogeneity. We study three specific constraints, limiting (1) the diameter of each cluster, (2) the sum of its branch lengths, or (3) chains of pairwise distances. These three problems can be solved in time that increases linearly with the size of the tree, and for two of the three criteria, the algorithms have been known in the theoretical computer scientist literature. We implement these algorithms in a tool called TreeCluster, which we test on three applications: OTU clustering for microbiome data, HIV transmission clustering, and divide-and-conquer multiple sequence alignment. We show that, by using tree-based distances, TreeCluster generates more internally consistent clusters than alternatives and improves the effectiveness of downstream applications. TreeCluster is available at https://github.com/niemasd/TreeCluster.

## Introduction

Homologous molecular sequences across different species or even within the same genome can show remarkable similarity due to their shared evolutionary history. These similarities have motivated many applications to first group the elements of a diverse set of sequences into *clusters* of set of sequences with high similarity for use in subsequent steps. The precise meaning of clusters depends on the application. For example, when analyzing 16S microbiome data, the standard pipeline is to use Operational Taxonomic Units (OTUs), which are essentially clusters of closely related sequences that do not diverge more than a certain threshold [1–3]. Another example is HIV transmission inference, a field in which a dominant approach is to

5P30AI027767-28 to M.B., U.M, N.M, and S.M. and NSF grant NSF-1815485 and NSF-1845967 to M.B., U.M., and S.M. Computations were performed on the San Diego Supercomputer Center (SDSC) through XSEDE allocations, which is supported by the NSF grant ACI-1053575. The funders had no role in study design, data collection and analysis, decision to publish, or preparation of the manuscript.

**Competing interests:** The authors have declared that no competing interests exist.

cluster HIV sequences from different individuals based on their similarity (again using a threshold) and to use these clusters as proxies to define clusters of disease transmission [4, 5].

Shared evolutionary histories, which is the origin of similarity among homologous sequences, can be shown using phylogenetic trees. The phylogenetic tree can be inferred from sequence data, [6, 7] and recently developed methods can infer approximate maximum-likelihood (ML) phylogenetic trees in sub-quadratic time, enabling them to scale to datasets of even millions of sequences [8]. Moreover, accurate alignment of datasets with hundreds of thousands of species (a prerequisite to most phylogenetic reconstruction methods) is now possible using divide-and-conquer methods [9, 10].

Most existing sequence clustering methods use the pairwise distances among sequences as input but do not take advantage of phylogenetic trees. For example, the widely-used UCLUST [2] searches for a clustering that minimizes the Hamming distance of sequences to the cluster centroid while maximizing the Hamming distance between centroids. Several other clustering methods have been developed for various contexts, such as gene family circumscription [11, 12] and large protein sequence databases [13].

Using phylogenies for clustering has two potential advantages. *i*) Since phylogenies explicitly seek to infer the evolutionary history, phylogeny-based clustering has the potential to not only reflect evolutionary distances (i.e., branch lengths) but also relationships (i.e., the tree topology). Recall also that branch lengths in a phylogeny are model-based "corrections" of sequence distances in a statistically-rigorous way [7, 14], and therefore, may better reflect divergence between organisms. *ii*) When inferred using subquadratic algorithms, the tree can eliminate the need to compute all pairwise distances, which can improve speed and scalability. Moreover, a phylogeny often has to be inferred for purposes other than clustering and thus typically is readily available. However, despite these potentials, to our knowledge, no systematic method for phylogeny-guided clustering exists. Built for analyzing HIV transmissions, ClusterPicker [15] clusters sequences based on their distances while using the phylogenetic tree as a constraint; however, it still uses sequence (not tree) distances and scales cubically with respect to number of sequences in the worst case.

Given a rooted phylogenetic tree, if the tree is ultrametric (that is, distances of all the leaves to the root are identical), clustering sequences based on the tree can proceed in an obvious fashion: the tree can be cut at some distance from the root, thereby partitioning the tree into clusters (Fig 1A). This approach extends in natural ways to unrooted ultrametric trees by first rooting the tree at the unique midpoint and proceeding as before. However, inferred phylogenetic trees are rarely ultrametric. Different organisms can evolve with different rates of evolution, and even when the rates are identical (leading to an ultrametric true tree), there is no guarantee that the inferred trees will be ultrametric. Given a non-ultrametric (and perhaps unrooted) tree, the best way to cluster sequences is not obvious (Fig 1B).

One way to approach tree-based clustering is to treat it as an optimization problem. We can define problems of the following form: "find the minimum number of clusters such that some criteria constrain each cluster." Interestingly, at least two forms of such optimization problems have been addressed as early as the 1970s by the theoretical computer science community, in the context of proving more challenging theorems. The *tree partitioning* problem is to cut a tree into the minimum number of subtrees such that the maximum path length between two nodes in the same subtree [17] or the sum of all edge weights in each subtree [18] is constrained by a given threshold. Both problems can be solved exactly using straightforward linear-time algorithms; however, to our knowledge, these algorithms are mostly ignored by bioinformaticians.

Here, we argue that a fast and efficient tree-based clustering approach can be beneficial to several questions in bioinformatics. In this paper, we introduce a family of tree partitioning

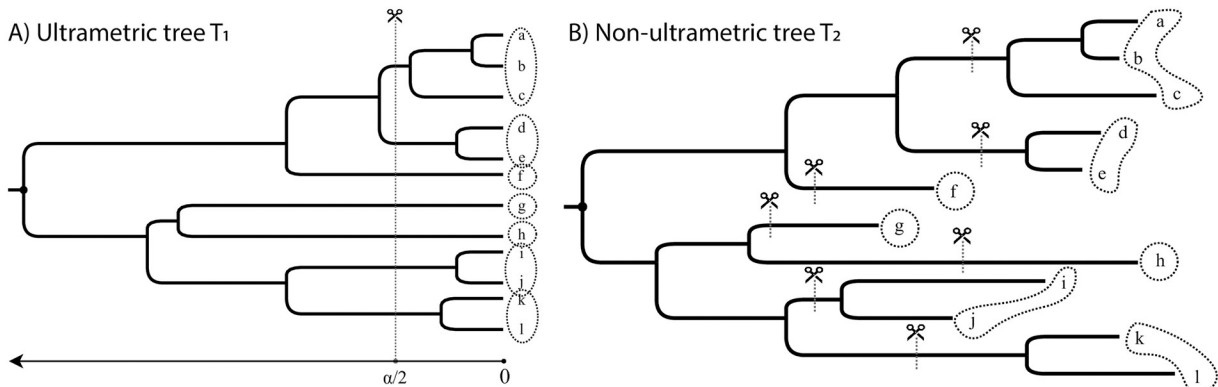

**Fig 1. When the phylogenetic tree is ultrametric, clustering is trivial.** For a threshold $\alpha$, cut the tree at $\frac{z}{2}$ height (A). When the tree is not ultrametric, it is not obvious how to cluster leaves (B). In both cases, a set of cut edges defines a clustering.

problems and describe linear-time solutions for three instances of the problem (two of which correspond to the aforementioned max and sum problems with known algorithms). We then show that tree-based clustering can result in improved downstream biological analyses in three different contexts: defining microbial OTUs, HIV transmission clustering, and divide-and-conquer multiple sequence alignment.

## Materials and methods

### Algorithms

**Problem definition.**   Let $T = (V, E)$ be an unrooted binary tree represented by an undirected acyclic graph with vertices $V$ (each with degree one or three), weighted edges $E$, and leafset $\mathcal{L} \subset V$. We denote the path length between leaves $u$ and $v$ on $T$ with $d_T(u, v)$ or simply $d(u, v)$ when clear by context. The weight of an edge $(u, v)$ (i.e., its branch length) is denoted by $w(u, v)$.

A clustering of the leaves of the tree $T$ can be defined by cutting a subset of edges $C \subseteq E$. We define a partition $\{L_1, L_2 \cdots, L_N\}$ of $\mathcal{L}$ to be an *admissible* clustering if it can be obtained by removing some edge set $C$ from $E$ and assigning leaves of each of the resulting connected components to a set $L_i$ (note: $N \leq |C| + 1$).

For a given tree $T$, let $f_T : 2^{\mathcal{L}} \to \mathbb{R}$ be a function that maps a subset of the leafset $\mathcal{L}$ to a real number. The purpose of $f_T$ is to characterize the diversity of elements at the leaves within each cluster, and it is often defined as a function of the edge weights in the cluster. For example, it can be the diameter of a subset: $f_T = \max_{u,v \in L} d_T(u, v)$. We define a family of problems that seek to minimize the number of clusters while each cluster has to adhere to constraints defined using $f_T$. More formally:

**Definition 1** (Min-cut partitioning problem family). *Given a tree $T$ with leafset $\mathcal{L}$ and a real number $\alpha$, find an admissible partition $\{L_1 \ldots L_N\}$ of $\mathcal{L}$ that satisfies $\forall i, f_T \leq \alpha$ and has the minimum cardinality ($N$) among all such clusterings.*

A natural way to limit the diversity within a cluster is to constrain all pairwise distances among members of the cluster to be less than a given threshold:

**Definition 2** (Max-diameter min-cut partitioning problem). *The Min-cut partitioning problem (Definition 1) is called Max-diameter min-cut partitioning problem when $f_T = \max_{u,v \in L} d(u, v)$.*

One potential disadvantage of max diameter min-cut partitioning is its susceptibility to outliers: the largest distance within a cluster may not be always an accurate representation of the

degree of diversity in the cluster. A natural choice that may confine the effect of outliers is the following:

**Definition 3** (Sum-length min-cut partitioning problem). *The Min-cut partitioning problem is called Sum-length min-cut partitioning problem when* $f_T = \sum\limits_{(u,v) \in edges(T|L)} w(u,v)$ *where* $T|L$ *is the tree T restricted to a subset of leaves L.*

We also study a third problem, which we will motivate later:

**Definition 4** (Single-linkage min-cut partitioning problem). *The Min-cut partitioning problem is called Single-linkage min-cut partitioning problem when* $f_T = \max\limits_{S \subset L} \{ \min\limits_{u \in S, v \in L-S} d(u,v) \}$.

Next, we will show linear-time algorithms for the Max-diameter, Sum-length, and Single-linkage min-cut partitioning problems. All three algorithms use variations of the same greedy algorithm and two of them (max and sum) have already been described in the theoretical computer science literature. Nevertheless, we reiterate the solutions using consistent terminology and provide alternative proofs of their correctness.

**Linear-time solution for Max-diameter min-cut partitioning.** A linear-time solution for the Max-diameter min-cut partitioning problem was first published by Parley *et al.* [17] (with all edge weights equal to 1). We present Algorithm 1, which is similar to the Parley *et al.* algorithm (but adds branch lengths), and we give an alternative proof. The algorithm operates on $T^o$, which is an arbitrary rooting of $T$ at node $o$. We denote the subtree rooted at an internal node $u$ as $U$. Let the two children of $u$ be called $u_l$ and $u_r$, and let the tree rooted by them be $U_l$ and $U_r$. We use $w_l$ and $w_r$ to denote $w(u, u_l)$ and $w(u, u_r)$, respectively, when clear by context.

**Algorithm 1**: Linear-time solution for Max-diameter min-cut partitioning

```
Input: A tree T° = (V, E) and a threshold α
1 B(v) ← 0 for v ∈ V
2 for u ∈ post order traversal of internal nodes of T° do
3   if B(uₗ) + wₗ + B(uᵣ) + wᵣ > α then
4     if B(uₗ) + wₗ ≤ B(uᵣ) + wᵣ then
5       E ← E - {(u, uᵣ)}
6       B(u) ← B(uₗ) + wₗ
7     else
8       E ← E - {(u, uₗ)}
9       B(u) ← B(uᵣ) + wᵣ
10   else
11     B(u) ← max(B(uₗ) + wₗ, B(uᵣ)+ wᵣ)
12 return Leafsets of every connected component in T°
```

For a cut set $C$ of the tree, we define $B(C, u)$ to be the length of the path from $u$ to the most distant *connected* leaf in $U$ in the clustering defined by $C$. The algorithm uses a bottom-up traversal of the tree and for each node $u$ that we visit, we may decide to cut one of its child edges. Thus, at each stage, a current clustering $C_u$ is defined; we use $B(u)$ a shorthand for $B(C_u, u)$. When we arrive at node $u$, one or more new paths form between the two trees $U_r$ and $U_l$. Among those paths, the longest one has the length $B(u_l) + w_l + B(u_r) + w_r$. If this value exceeds the threshold, we break either $(u, u_r)$ or $(u, u_l)$, depending on which minimizes $B(u)$. Note that the algorithm always cuts at most one child edge of every node, and thus, $B(u)$ is always well-defined.

**Theorem 1**. *Let $A(u)$ be the minimum number of clusters under $U$, each with a diameter less than $\alpha$ (i.e., $A(o)$ is the objective function). Algorithm 1 computes a clustering with the minimum $A(o)$ for the rooted tree $T^o$. In addition, among all possible such clusterings, the algorithm picks $\arg\min_C B(C, o)$.*

**Corollary 1**. *Let $C'$ be the cut set obtained by running Algorithm 1 on an arbitrary rooting $T^o$ of tree $T$. $C'$ optimally solves the Max-diameter min-cut partitioning problem.*

The proof of the theorem and the corollary are both given in S1 Appendix.

**Linear solution for the Sum-length min-cut partitioning problem.**   A linear-time algorithm that partitions trees into the fewest clusters, each with total node weights less than or equal to $\alpha$, has been previously published by Kundu *et al.* [18]. In order to solve the Sum-length min-cut partitioning problem, we present an altered version of the original algorithm that works on edge (instead of node) weights and that focuses on binary trees. Algorithm 1 with two simple modifications solves the Sum-length min-cut partitioning problem optimally (see Algorithm A in S1 Appendix). The first modification is that we define the auxiliary variable $B(C, u)$ to denote the sum of weights of all descendent edges connected to $u$ at the stage it is processed by the algorithm. Secondly, in the bottom-up traversal of internal nodes of $T^o$, for node $u$, w.l.o.g, let $B(u_l) + w_l \geq B(u_r) + w_r$. If the sum of branch lengths in the combined subtree exceeds $\alpha$, we break the edge $(u, u_l)$. Unlike Algorithm 1, where $B(u_l) + w_l + B(u_r) + w_r \leq \alpha$, here, $B(u)$ is set to $B(u_l) + w_l + B(u_r) + w_r$. The proof for the correctness of the algorithm is analogous to that of Algorithm 1 and is given in S1 Appendix.

**Single-linkage min-cut partitioning.**   We now address the Single-linkage min-cut partitioning problem (Definition 4), which can be considered a relaxation of the Max-diameter min-cut partitioning. To motivate this problem, first consider the following definition.

**Definition 5** (Single-linkage clustering). *We call a partition of $\mathcal{L}$ to be a Single-linkage clustering when for every $a, b \in \mathcal{L}$, a and b are in the same cluster if and only if there exists a chain $\mathcal{H} = c_0, c_1, \ldots, c_m, c_{m+1}$, where $a = c_0$ and $b = c_{m+1}$, and for every $0 \leq i \leq m$, we have $d(c_i, c_{i+1}) \leq \alpha$.*

Thus, every pair of nodes is put in the same cluster if (but not only if) their distance is below the threshold (the rest follows from transitivity). The next result (proved in S1 Appendix.) motivates the choice of $f_T$ in Definition 4.

**Proposition 1**. *The optimal solution to the Single-linkage min-cut partitioning problem (Definition 4) is identical to the Single-linkage clustering of Definition 5.*

Algorithm 2 shows a linear-time solution to the Single-linkage min-cut partitioning problem. For each node $u$, the algorithm first finds the closest leaf in the left and right sub-trees of $u$ via post-order traversal, and it then finds the closest leaf outside the sub-tree rooted at $u$ via pre-order traversal. Then, on a post-order traversal, it cuts each child edge iff the minimum distance of leaves under it to leaves under its sibling *and* to any leaf outside the node both exceed the threshold. The following theorem states the correctness of the algorithm (proof is given in S1 Appendix).

**Algorithm 2**: Sᴵɴɢʟᴇ-Lɪɴᴋᴀɢᴇ Single-linkage min-cut partitioning

```
1  minBelow[u] ← minAbove[u] ← ∞ for v ← V
2  for u∈ post order traversal of Tᴼ do
3    if u in 𝓛 then
4      minBelow[u] ← 0;
5    else
6      minBelow[u] ← min(minBelow[uₗ] + wₗ, minBelow[uᵣ] + wᵣ);
7  for u ∈ pre order traversal of Tᴼ do
8    if u ≠ o then
9      minAbove[u] ← min(minBelow[s] + w(v, s), minAbove[v] + w(v, v));
10 for u ∈ post order traversal of internal nodes of Tᴼ do
11   if minBelow[uₗ] + wₗ + minBelow[uᵣ] + wᵣ > α and
      minBelow[uₗ] + wₗ + minAbove[u] > α then
12     E ← E\(u, uₗ)
13   if minBelow[uₗ] + wₗ + minBelow[uᵣ] + wᵣ > α and
      minBelow[uᵣ] + wᵣ + minAbove[u] > α then
14     E ← E\(u, uᵣ)
15   if minBelow[uₗ] + wₗ + minAbove[u] > α and
      minBelow[uᵣ] + wᵣ + minAbove[u] > α then
```

```
16      E ← E\(v, u)
17 return Leafsets of every connected component in T^O
```

**Theorem 2**. *The partitioning computed by Algorithm 2 optimally the solves Single-linkage min-cut partitioning problem* (*Definition 5*).

**Clade constraint for rooted trees.** So far, we have focused on unrooted trees. This choice is partially driven by the fact that phylogenetic reconstruction tools predominantly use time-reversible models of sequence evolution (e.g. GTR [19]) and therefore output an unrooted tree. Nevertheless, researchers have developed various methods for rooting trees [20, 21], including accurate and linear-time methods such as MV rooting [16]. When a rooted tree is available, each "monophyletic clade," i.e., group of entities that includes all descendants of their common ancestor, is a biologically meaningful unit. Thus, we may want to constrain each cluster to be a clade. These "clade" constraints make clustering easier: our algorithms can be easily altered to ascertain that each cluster is also a clade. Specifically, in Algorithm 1, when we have $B(u_l) + w_l + B(u_r) + w_r > \alpha$, we simply need to cut both $(u, u_l)$ and $(u, u_r)$ (instead of cutting only the longer one). This small modification allows the Max-diameter, Sum-length, and Single-linkage min-cut partitioning problems to be solved in linear time while imposing the clade constraint.

**Centroid (representative) sequence.** Many sequencing clustering methods produce a representative sequence per cluster, often one that is used internally by the algorithm. Our clustering approach is representative-free. However, if a representative is needed for down-stream applications, several choices are available. For example, one can in linear-time find the midpoint or balance point of a cluster [16] (i.e., the node that minimizes variance of root to tip distances); then, the leaf closest to the midpoint or balance point can be used as the representative. Another alternative is to use the consensus sequence among all sequences belonging to a cluster (i.e., choosing the most frequent letter for each site). Constructing and using a consensus sequence may be preferable to using one of the given sequences as the centroid [22]. A third alternative that we explore in our results is to use ancestral sequence reconstruction. For each subtree defined by a cluster, we first root it at its balance point. Then, we perform maximum likelihood ancestral state reconstruction (ASR) and use the reconstructed root sequence as the centroid.

## TreeCluster software

We implemented linear-time algorithms for min-cut partitioning problem subject to Max-diameter, Sum-branch, Single-linkage, and other clustering criteria, with and without clade constraints in a freely-available open source tool called TreeCluster. TreeCluster takes a newick tree and a threshold value as input and returns clusters in a formatted text file. TreeCluster uses treeswift [23] package for fast tree operations.

## Three applications of TreeCluster

While sequence clustering has many applications, in this paper, we highlight three specific areas as examples.

**Application 1: OTU clustering. Biological Problem**. For microbiome analyses using 16S sequences generated from whole communities, the standard pipeline uses operational taxonomic units (OTUs). Sequences with similarity at or above a certain threshold (e.g. 97%) are grouped into OTUs, which are the most fine-grained level at which organisms are distinguished. All sequences assigned to the same OTU are treated as one organism in downstream analyses, such as taxonomic profiling, taxonomic identification, sample differentiation, or machine learning. The use of a similarity threshold instead of a biological concept of species is

to avoid the notoriously difficult problem of defining species for microbial organisms [24, 25]. Futher, the use of clusters of similar sequences as OTUs can provide a level of robustness with respect to sequencing errors.

Most applications of OTUs are closed-reference: a reference database of known organisms is selected, and OTUs are defined for reference sequences using methods such as UCLUST [2] and Dotur [3]. These methods cluster sequences based on a chosen threshold of similarity, often picking a centroid sequence to represent an OTU. Reads from a 16S sample are then compared to the OTUs, and the closest OTU is found for each read (judging distance by sequence similarity). Once all reads are processed for all samples, an OTU table can be built such that rows represent samples, columns represent OTUs, and each cell gives the frequency of an OTU in a sample. This table is then used in downstream analyses. Several large reference databases exist for these OTU-based analyses [26–28]. One of these databases, popularized through pipelines such as Qiita [29], is Greengenes [28].

Regardless of the downstream application of an OTU table, one would prefer the OTUs to be maximally coherent (i.e., internally consistent) so they represent organisms as faithfully as possible. We will focus our experiments on the closed-reference OTU picking methods and the Greengenes as the reference library. However, note that open-reference OTU picking and sub-operational-taxonomic-unit (sOTU) methods [30–32] also exist and involve a similar need for sequence clustering.

**Existing methods**. Despite the availability of hierarchical clustering tools for OTU clustering [3, 33], non-hierarchical clustering methods [2, 34] are more widely used, perhaps due to their lower computational demand. Two prominent methods are UCLUST [2] and CD-HIT [34], which share the same algorithmic strategy: for a given threshold $\alpha$, UCLUST determines a set of representative sequences dynamically by assigning query sequences into representative sequences (centroids) such that, ideally, the distance between each query and its assigned centroid is less than $\alpha$ while distances between centroids is more than $\alpha$. UCLUST is a heuristic algorithm, and the processing order of the queries may affect the resulting clustering. CD-HIT differs from UCLUST primarily in its strategy for computing distances.

**Formulation as min-cut partitioning**. We define OTUs by solving the Min-diameter, Sum-Length, or Single-linkage min-cut partitioning problems using a chosen threshold $\alpha$ and an inferred ML phylogeny. Each cluster in the resulting partition is designated as an OTU.

**Experiments**. We evaluate the quality of tree-based OTU clustering by comparing it to UCLUST as used by Greengenes [28]. We run TreeCluster on the phylogenetic tree of 203,452 sequences in the Greengenes v13.5 database in three modes: max, sum, and single-linkage. We use the following 20 thresholds: [0.005, 0.05] with a step size of 0.005, and (0.05, 0.15] with a step size of 0.01. For single-linkage, we only go up to 0.1 because, above this threshold, the number of clusters becomes much smaller than other methods.

From the same Greengenes database, we extract OTU clusters for all available sequence identity thresholds up to 0.15 (i.e., 0.03, 0.06, 0.09, 0.12, and 0.15). We measure the quality of a clustering $\{L_1, \ldots, L_N\}$ by its weighted average of average pairwise distance per cluster (which we call *cluster diversity* for shorthand), given by the following formula:

$$\mu(\{L_1, \ldots, L_N\}) = \frac{\sum_{k=1}^{N} |L_k| \frac{\sum_{i,j \in L_k} d(i,j)}{|L_k|^2}}{\sum_{k=1}^{N} |L_k|} = \frac{1}{n} \sum_{k=1}^{N} \sum_{i,j \in L_k} \frac{d(i,j)}{|L_k|} \tag{1}$$

where $n$ denotes the number of sequences clustered. We compute distance $d(i, j)$ between two elements using two methods: *tree distance*, which is the path length on the inferred

phylogenetic tree, and sequence-based Hamming distance. Hamming distances are computed pairwise from the multiple sequence alignment of all 203,452 sequences in the Greengenes database and ignore any site that includes a gap in the pairwise alignment. Clearly, cluster diversity alone is insufficient to judge results (singletons have zero diversity). Instead, we compare methods at the same level of clustering with respect to their diversity. Thus, as we change the threshold $\alpha$, we compare methods for choices of the threshold where they result in (roughly) equal numbers of clusters. Given the same number of clusters, a method with lower cluster diversity is considered preferable.

We measure the quality of a representative sequence set using two metrics. For a clustering $\{L_1, \ldots, L_N\}$, let $\{L_1, \ldots, L_{N'}\}$ denote all non-singleton clusters. The first metric is the average of average distance to the centroid per cluster, formally defined as:

$$v(g, \{L_1, \ldots, L_{N'}\}) = \frac{1}{N'} \sum_{k=1}^{N'} \sum_{i \in L_k} \frac{d(i, g(L_k))}{|L_k|} \tag{2}$$

where $g$ is a function that maps a cluster to a (representative) sequence. The second metric is the average of maximum distance to the representative per cluster, formally defined as:

$$\xi(g, \{L_1, \ldots, L_{N'}\}) = \frac{1}{N'} \sum_{k=1}^{N'} \max_{i \in L_k} d(i, g(L_k)) \tag{3}$$

We define these metrics on the set of non-singleton clusters because a trivial clustering which assigns many singletons will trivially have a very low value for $v$ and $\xi$ (near zero).

Greengenes database is distributed with precomputed representative sequence sets. For centroid selection for TreeCluster, we consider two methods $g$: consensus and ASR. We perform ASR using TreeTime [35] under GTR model. We use RAxML 8 [36] to infer GTR model parameters from the Greengenes multiple sequence alignment of representative sequences at 15 percent threshold. We compute distance $d(i, j)$ between two elements using Hamming distance.

**Application 2: HIV transmission cluster analyses.   Biological Problem**. HIV evolves rapidly, so phylogenetic relationships between sequences contain information about the history of transmission [37]. The ability to perform phylogenetic analyses of HIV sequences is critical for epidemiologists who design and evaluate HIV control strategies [38–42]. The results of these analyses can provide information about the genetic linkage [43] and transmission histories [44], as well as mixing across subpopulations [45]. A recent advancement in computational molecular epidemiology is the use of transmission clustering to predict at-risk individuals and epidemic growth: infer transmission clusters from pairwise sequence distances, monitor the growth of clusters over time, and prioritize clusters with the highest growth rates [46]. In this monitoring framework, two natural questions come about: What is the optimal way to infer transmission clusters from molecular data, and how can transmission cluster inference be performed more efficiently?

**Existing methods**. We focus on two popular tools that perform such clustering. Cluster Picker [4] is given a distance threshold, a phylogenetic tree, and sequences. It clusters individuals such that each cluster defines the leaves of a clade in the tree, the maximum pairwise sequence-based distance in each cluster is below the threshold, and the number of clusters is minimized. HIV-TRACE is a tool that, given a distance threshold and sequences, clusters individuals such that, for each pair of individuals $u$ and $v$, if the Tamura-Nei 93 (TN93) distance [47] between $u$ and $v$ is below the threshold, $u$ and $v$ are placed in the same cluster [5]. Both methods scale worse than linearly with the number of sequences (quadratically and cubically,

respectively, for HIV-TRACE and Cluster Picker), and for large datasets, they can take hours, or even days, to run (however, HIV-TRACE enjoys trivial parallelism and is fast in practice).

**Formulation as min-cut partitioning**. Transmission clustering is similar to our problem formulation in that it involves cutting edges such that the resulting clusters (as defined by the leafsets resulting from the cuts) must adhere to certain constraints. Both Cluster Picker and HIV-TRACE utilize pairwise distances computed from sequences, but when reformulated to utilize tree-based distances from an inferred phylogeny, Cluster Picker becomes analogous to our Max-diameter min-cut partitioning (with an added constraint that clusters must define clades in the phylogeny), and HIV-TRACE becomes analogous to the Single-linkage min-cut partitioning.

**Experiments**. To evaluate the effectiveness of HIV transmission clustering, we first simulate HIV epidemic data using FAVITES [48]. For the simulation parameters, we use the parameters described in Moshiri *et. al.* [48] to model the San Diego HIV epidemic between 2005 and 2014. However, we deviate from the original parameter set in one key way: originally, all HIV patients were sequenced at the end time of the epidemic, yielding an ultrametric tree in the unit of time, but to better capture reality, we instead sequence each patient the first time they receive Antiretroviral Treatment (ART). In our simulations, we vary two parameters: the expected time to begin ART as well as the expected degree of the social contact network, which underlies the transmission network. Higher ART rates and lower degrees both result in a slower epidemic and change patterns of phylogenetic branch length [48]. The complete FAVITES parameter set can be found in the supplementary materials (List A in S1 Appendix). We infer phylogenies from simulated sequences under the GTR+Γ model using FastTree-II [8], and we use the MinVar algorithm to root the trees using FastRoot [49].

We use HIV-TRACE [5] as well as multiple clustering modes of TreeCluster to infer transmission clusters. We were unable to use Cluster Picker [4] due to its excessive running time. For HIV-TRACE, we use a clustering threshold of 1.5% as suggested by its authors [46]. Because HIV-TRACE estimates pairwise sequences distances under the TN93 model, [47] which tend to be underestimates of phylogenetic distance estimated under the GTR model, we use a clustering threshold of 3% for Single-Linkage TreeCluster. The default Cluster Picker threshold for Max-diameter clustering is 4.5% [4], so we use this as our clustering threshold for Max-Diameter TreeCluster (both with and without the Clade constraint). For Sum-length TreeCluster (with and without the Clade constraint), we simply double the Max-diameter threshold and use 9%. In addition to using these default thresholds, we also test a wide range of thresholds for each transmission clustering method for robustness.

We measure cluster growth from year 8 to year 9 of the simulation and select the 1,000 highest-priority individuals, where individuals are prioritized in descending order of respective cluster growth. To measure the risk of a given individual $u$, we count the number of HIV transmission events $u \to v$ between years 9 and 10. To measure the effectiveness of a given clustering, we average the risk of the selected top 1,000 individuals. Higher numbers imply the ability to prevent more transmissions by targeting a fixed number of individuals (1,000) and are thus desirable. As a control, we also show the mean number of transmissions per population, which is what a random selection of 1,000 individuals would give in expectation (we call this "expected" risk).

**Application 3: Divide-and-conquer multiple sequence alignment.** **Algorithmic idea**. Tree-based clustering has also been used for multiple sequence alignment (MSA) using divide-and-conquer. To solve the MSA problem using divide-and-conquer, the tree structure can be used to divide sequences into smaller subsets (i.e., clusters), which can each be aligned separately and then merged. The phylogeny and the MSA can be inferred simultaneously by iterating between tree and MSA inference, and this technique has been used in algorithms such as

SATe [50, 51] and PASTA [52]. Divide-and-conquer has been proven to be particularly useful for MSA of very large datasets [9, 10, 50]. We note that not all MSA tools use divide-and-conquer and that we only study the usage of min-cut partitioning in divide-and-conquer methods. We examine the effectiveness of min-cut partitioning in PASTA [52], a scalable software which infers both MSAs and trees for ultra-large datasets (tested for up to 1,000,000 sequences).

PASTA first builds a quick-and-dirty estimate of the phylogeny that is used as a guidance to cluster the sequences. In its "divide" phase, PASTA clusters the input sequences into subsets so that each subset contains less diverse sequences than the full set. Then, an accurate (but often computationally demanding) method is run on the subsets to infer the MSA and/or the tree. Finally, the results on the subsets are merged using various techniques. The accuracy of the output depends not only on the accuracy of the base method used on the subsets and the merging method, but also on the effectiveness of the method used to divide the tree into subsets [51].

PASTA computes an initial alignment using HMMs implemented in HMMER [53] and an initial tree using FastTree-II [8]; then, it performs several iterations (3 by default) of the divide-and-conquer strategy described before using MAFFT [54] for aligning subsets and using a combination of OPAL [55] and a technique using transitivity for merging subalignments. A tree is generated using FastTree-II at the end of each iteration, which is then used as the guide tree for the next iteration. The method has shown great accuracy on simulated and real data, especially in terms of tree accuracy, where it comes very close to the accuracy obtained using the true alignment, leaving little room for improvement. However, in terms of the alignment accuracy, it has substantial room for improvement on the most challenging datasets.

The clustering used in PASTA is based on the centroid-edge decomposition. Given the *guide tree* (available from the previous iteration), the decomposition is defined recursively: divide the tree into two halves, such that the two parts have equal size (or are as close in size as possible). Then, recurse on each subtree until there are no more than a given number of leaves (200 by default) in each subset.

**Formulation as min-cut partitioning**. The centroid edge decomposition involves cutting edges and includes a constraint defined on the subsets. However, it is defined procedurally and does not optimize any natural objective function. The min-cut partitioning can produce a decomposition similar to the centroid decomposition in its constraints but different in outcome. We set all edge weights of the guide tree to 1 and solve the Sum-length min-cut partitioning problem with threshold $\alpha = 2m - 2$; the result is a partition such that no cluster has more than $m$ leaves and the number of subsets is minimized. Thus, this "max-size min-cut partitioning" is identical to centroid decomposition in its constraints but guarantees to find the minimum number of clusters.

**Experiments**. To evaluate how our new decomposition impacts PASTA, we run PASTA version 1.8.3 on two datasets, and for each, we compare the accuracy of the two decomposition strategies: centroid and max-size min-cut partitioning. Other parameters (including maximum subset size) are all kept fixed for both decomposition strategies. We used two datasets both from the original PASTA paper: 10 replicates of a simulated RNAsim dataset with 10,000 leaves and a set of 19 real HomFam datasets with 10,099 to 93,681 protein sequences. The RNASim is based on a very complex model of RNA evolution. Here, the true alignment, known in simulations, is used as the reference. For HomFam, since the true alignment is not known, following previous papers, we rely on a very small number of seed sequences with a hand-curated reliable alignment as reference [9, 56]. In both cases, we measure alignment error using two standard metrics computed using FastSP [57]: SPFN (the percentage of

homologies in the reference alignment not recovered in the estimated alignment) and SPFP (the percentage of homologies in the estimated alignment not present in the reference).

## Results

### Results for Application 1: OTU clustering

On the Greengenes dataset, as we change the threshold between 0.005 and 0.15, we get between 181, 574 and 10, 112 clusters (note that singletons are also counted). The cluster diversity has a non-linear relationship with the number of clusters: it drops more quickly with higher thresholds where fewer clusters are formed (Fig 2A and S1 Fig). Comparing the three objective

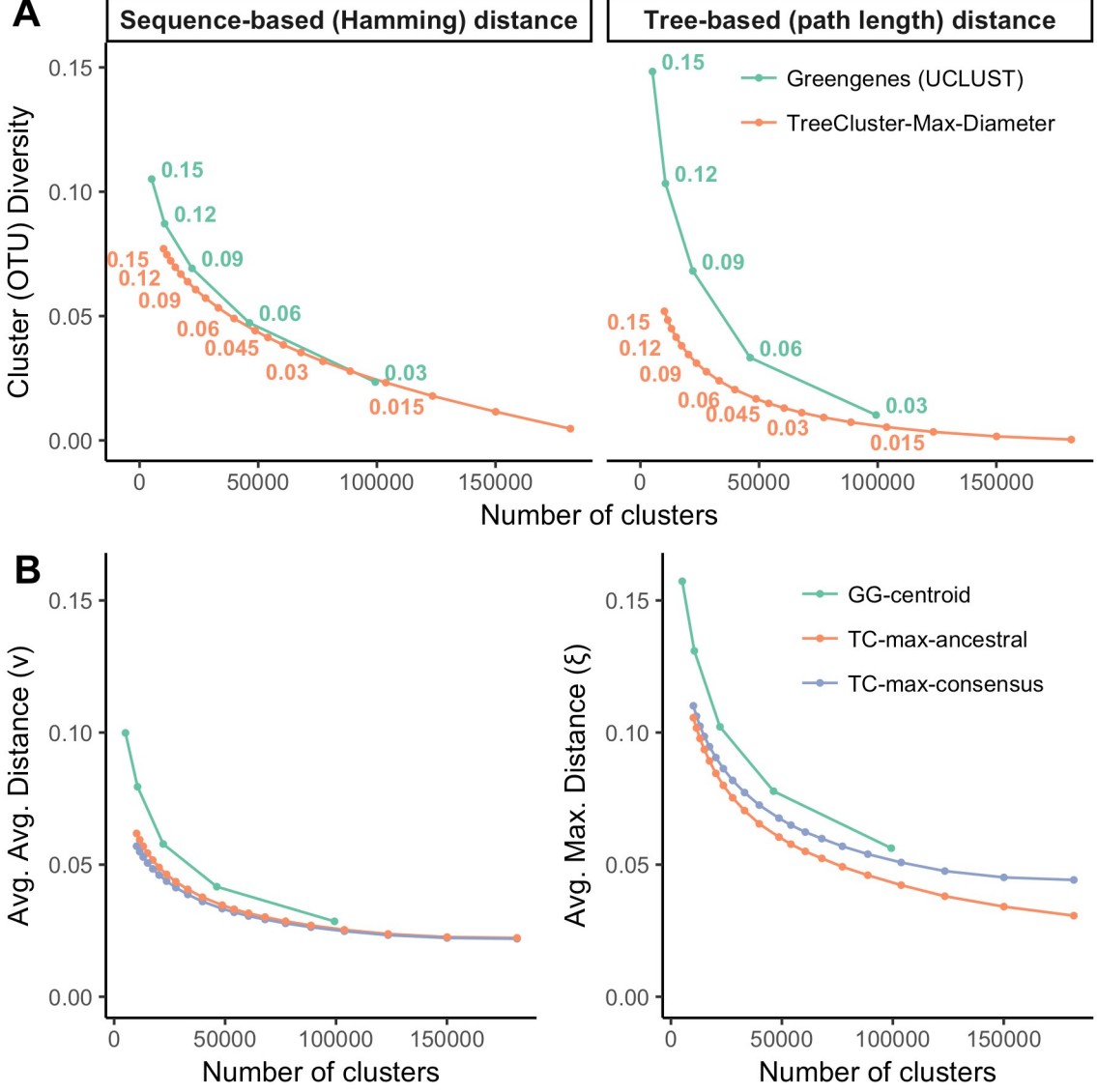

**Fig 2. Comparing Greengenes and TreeCluster.** (A) Cluster diversity (Eq 1) for Greengenes and TreeCluster versus the number of OTUs. Cluster diversity is measured both with respect to hamming distance and tree-based distance. The threshold $\alpha$ is shown for all data points corresponding to Greengenes and for some points of TreeCluster. See S1 Fig for comparison to other TreeCluster modes. (B) Average-average ($v$) and average-maximum ($\xi$) distance to the centroid for Greengenes and TreeCluster versus the number of clusters. TreeCluster centroids are computed using ancestral state reconstruction or using consensus.

functions that can be used in TreeCluster, we observe that Max-diameter and Sum-length have similar trends of cluster diversity scores, whereas Single-linkage min-cut partitioning has substantially higher diversity compared to the other two methods (S1 Fig). This pattern is observed regardless of whether tree distances or sequence distances are used, but differences are larger for tree distances. Finally, note that, even though tree distances are, as expected, larger than sequence distances (S2 Fig), the cluster diversity is *lower* when computed using tree distances, showing that clusters are tight in the phylogenetic space.

Compared to the default Greengenes OTUs, which are defined using UCLUST, Max-diameter min-cut partitioning defines tighter clusters for tree-based scores (Fig 2A). When distances between sequences are measured in tree distance, the cluster diversity score for Greengenes OTUs is substantially lower for all thresholds, and the gap is larger for higher thresholds. For example, the cluster diversity of Greengenes OTUs is three times higher than TreeCluster OTUs for $\alpha$ = 0.15. When distances between sequences are measured in Hamming distance, Greengenes and TreeCluster perform similarly for low threshold values (e.g. $\alpha$ = 0.03 for Greengenes, which is similar to $\alpha$ = 0.02 for TreeCluster in terms of the number of clusters). However, when the number of OTUs is reduced, remarkably, TreeCluster outperforms Greengenes OTUs by up to 1.4-fold (e.g. $\alpha$ = 0.15). This is despite the fact that UCLUST is working based on sequence distances and TreeCluster is not.

Size of the largest cluster in Greengenes is larger compared to TreeCluster (Table 1). For example, for $\alpha$ = 0.09, both methods have similar number of clusters (22,090 and 23,631 for Greengenes and TreeCluster, respectively) but the size of largest cluster in Greengenes is three times that of TreeCluster (1,659 versus 540). On the other hand, for the same threshold value, the number of singleton clusters comprises 48% of all clusters for Greengenes whereas only 27% of the clusters are singletons for TreeCluster. Thus, GreenGenes has more clusters that are very small or very large, compared to TreeCluster.

Computed using either consensus or ASR method, representative sequences in TreeCluster are closer to other sequences of the cluster than Greengenes (Fig 2B). Using ASR representative sequences performs slightly worse than consensus centroids according to the $\nu$ score (e.g. $\nu$ = 0.062 and $\nu$ = 0.057, respectively when $\alpha$ = 0.15). When evaluated using $\xi$ score, ASR representative sequences perform slightly better than consensus in all threshold levels (e.g. $\xi$ = 0.03 and $\xi$ = 0.04 respectively when $\alpha$ = 0.005) and the gap again widens as the number of clusters increases. Both types of centroids computed using TreeCluster perform better than Greengenes representative sequences according to both metrics, and the gap increases as the threshold $\alpha$ increases (e.g. up to 1.7-fold when $\alpha$ = 0.15 for $\nu$).

## Results for Application 2: HIV dynamics

Comparing various TreeCluster modes, regardless of the parameters that we vary, Sum-length TreeCluster consistently outperforms the other clustering methods, and the inclusion of the Clade constraint has little impact on effectiveness (Fig 3). Compared to a random selection of individuals, the risk of selected individuals can be substantially higher; for example, with expected time to begin ART set to 1 year, the expected risk is 0.55 transmissions, whereas the average risk of the top 1,000 individuals from Sum-length clusters is 0.85. In all the conditions, a close second to TreeCluster Sum-length is TreeCluster Max-diameter. Other methods, however, are substantially less effective than these two modes of TreeCluster.

When varying expected time to begin ART and expected degree, Single-linkage TreeCluster and HIV-TRACE consistently perform lower than the other approaches, with Single-linkage TreeCluster typically performing around the theoretical expectation of a random selection and HIV-TRACE performing slightly better (Fig 3a and 3b). Moreover, these

**Table 1. Number of singleton clusters (σ), total number of clusters (Σ), and maximum cluster size (max) for TreeCluster and GreenGenes for various thresholds.** In the Greengenes database, OTU definitions for thresholds α = 0.015 and α = 0.045 are not available.

| | TreeCluster-Max-Diameter | | | GreenGenes-UCLUST | | |
|---|---|---|---|---|---|---|
| α | σ | Σ | max | σ | Σ | max |
| 0.015 | 86387 | 123456 | 47 | (n.a) | (n.a) | (n.a) |
| 0.03 | 42510 | 77263 | 96 | 70415 | 99322 | 527 |
| 0.045 | 24795 | 54068 | 171 | (n.a) | (n.a) | (n.a) |
| 0.06 | 15257 | 39809 | 305 | 26485 | 46256 | 894 |
| 0.09 | 6396 | 23631 | 540 | 10560 | 22090 | 1659 |
| 0.12 | 3003 | 15052 | 808 | 4153 | 10544 | 2131 |
| 0.15 | 1525 | 10112 | 1209 | 1735 | 5088 | 3765 |

patterns are not simply due to the chosen thresholds: even when the threshold is changed to control the number of clusters, Single-linkage TreeCluster and HIV-TRACE consistently perform worse than expected by random selection (Fig 3c). The effectiveness of Sum-length and Max-diameter TreeCluster are maximized when they create 2,000–5,000 and 2,000–3,000 clusters, respectively.

## Results for Application 3: Improving PASTA

First, we notice a substantial improvement of PASTA 1.8.3 comparing to the original version published in 2015 [52], especially for RNASim where alignment accuracy improved by about 3%. This was due to major updates of the PASTA software and the dependent tools. When we replace centroid decomposition with Max-size min-cut partitioning in PASTA, the alignment error reduces substantially for the RNASim dataset, but less so on the HomFam dataset (Fig 4). On the RNASim data, mean SPFN drops from 0.12 to 0.10, which corresponds to a 17% reduction in error. These drops are consistent across replicates and are substantial given the fact that the only change in PASTA was to replace its decomposition step with our new clustering algorithm, keeping the rest of the complex pipeline unchanged. In particular, the method to align subsets, to merge alignments, and to infer trees, were all kept fixed. On the HomFam dataset, too, errors decreased, but the reductions were not substantial (Fig 4b). Based on these results, we have now changed PASTA to use Max-size min-cut partitioning by default.

## Discussion

Several theoretical and practical issues should be further discussed.

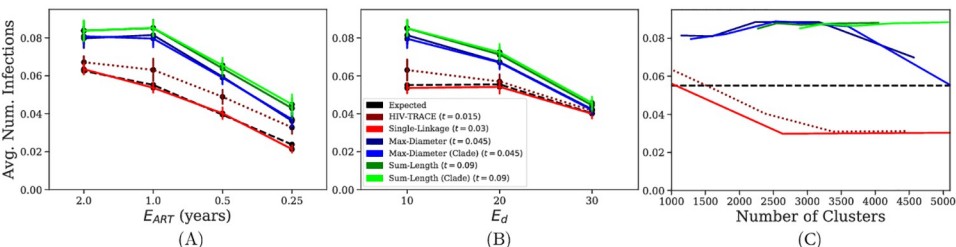

**Fig 3. Effectiveness of transmission clustering.** Effectiveness is measured as the average number of individuals infected by the selected 1,000 individuals. The horizontal axis depicts the expected time to begin ART (A), the expected degree (i.e., number of sexual contacts) for individuals in the contact network (B), and the number of clusters using various thresholds (C).

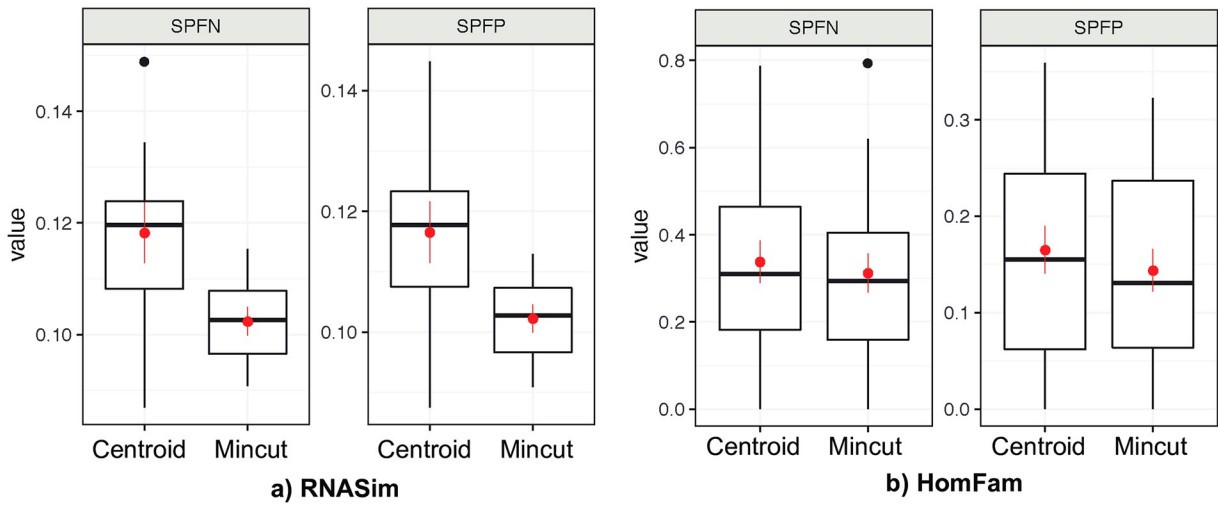

**Fig 4. Alignment error for PASTA using the centroid and the mincut decompositions.** We show Sum of Pairs False Negative (SPFN) and Sum of Pairs False Psotive (SPFP) computed using FastSP [57] over two datasetes: the simulated RNASim dataset (10 replicates) and the biological HomFam dataset (19 largest families; all 20 largest, except "rhv" omitted due to the warning on the Pfam website). We show boxplots in addition to mean (red dot) and standard error (red error bars).

## Mean-diameter min-cut partitioning

Some of the existing methods, such as Cluster Picker [4], can define their constraints based on mean pairwise distance between nodes. Similar to those, we can define a variation of the min-cut partitioning problem in which $f_T(L) = \frac{1}{\binom{|L|}{2}} \sum_{i,j \in L} d(i,j)$. Unfortunately, this "Mean-diameter" min-cut partitioning problem can only be solved in linear time using our greedy algorithm if we also have clade constraints (Algorithm B in S1 Appendix). As demonstrated by the counterexample given in S3 Fig, the greedy algorithm fails if we do not have clade constraints. More generally, the use of mean as function $f_T(\cdot)$ creates additional complexity, and whether it can be solved in linear time remains unclear. Whether mean diameter is in fact a reasonable criterion is not clear. For example, it is possible that the mean diameter of a cluster is below the threshold while the mean diameter of subclusters embedded in that cluster are not; such scenarios may not make sense for downstream applications.

## Set of optimal solutions

It is possible that multiple distinct partitions with equal number of clusters are all optimal solutions to any of our min-cut partitioning problems. Moreover, as the example given in Fig 5 shows, the number of optimal solutions can be exponential with respect to number of leaves in a binary phylogenetic tree. This observation renders listing all optimal solutions potentially impractical as there may be too many of them. However, finding a way to summarize all optimal partitions remains interesting and can have practical utility. We do not currently have such a summarization approach. However, as shown in Lemma A of S1 Appendix, although the optimal solution space is potentially exponentially large, one can easily determine the set of all edges that could appear in any of the optimal solutions. Thus, we could find absolutely unbreakable edges that will not be cut in any optimal clustering of the data.

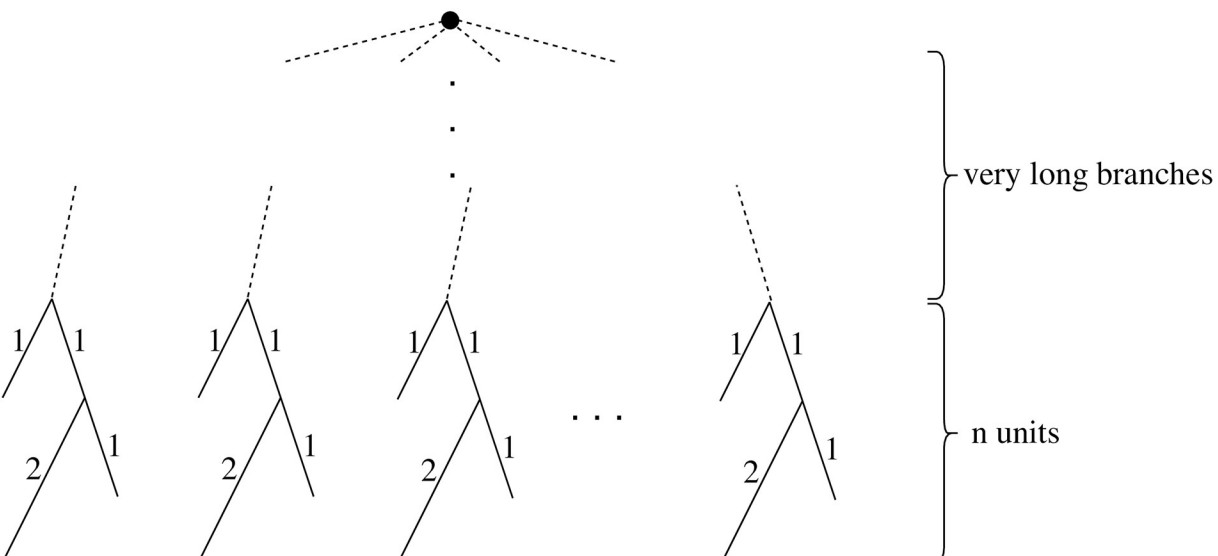

**Fig 5. An example showing that number of minimal clusterings under a diameter threshold can be exponential of number of leaves.** When the threshold is 3.5, each unit has to be split into two clusters, and there are thus three equally-optimal ways of splitting. The minimum number of clusters is therefore $2n$. The total number of distinct optimal solutions is $3^n$, whereas there are $3n$ leaves.

### Choice of criterion

Among the three methods that we discussed, we observed that Max-diameter and Sum-diameter are consistently better than the Single-linkage. This observation makes intuitive sense. Single-linkage can increase the diversity within a cluster simply due to the transitive nature of its criterion. Thus, a very heterogeneous dataset may still be collapsed into one cluster, simply due to transitivity. Our desire to solve the Single-linkage problem was driven by the fact that a similar concept is used in HIV-TRACE, arguably the most widely used HIV clustering method. However, we did not detect any advantage in this type of clustering compared to Max-diameter or Sum-length; thus, our recommendation is to use these two criteria instead. Between the two, Max-diameter has the advantage that its $\alpha$ threshold is easier to interpret. Finally, ASR-based selection of representative sequences outperformed consensus sequences, but we note that computing consensus sequences is much easier and faster.

### Running time

We focused on comparing the effectiveness of TreeCluster to other methods, but we note that its running time also compares favorably to other clustering methods (once the tree is inferred). For example, on a real HIV dataset, we ran HIV-TRACE, Cluster Picker, and TreeCluster for subsets of the data ranging from 100 to 5,000 sequences (Fig 6). On the largest set with 5,000 leaves, the running time of TreeCluster did not exceed 2 seconds. In contrast, the sequence-based HIV-Trace required close to a minute (which is still quite fast), but Cluster Picker needed more than an hour. Even on the Greengenes dataset with more than 200,000 leaves, TreeCluster performed clustering in only 30 seconds. The high speed of TreeCluster makes it possible to quickly scan through a set of $\alpha$ thresholds to study its impact on the outcomes of downstream applications.

We note that these numbers do not include the time spent for inferring the tree, which should also be considered if the tree is not already available (note that in many applications a tree is inferred for other purposes and is readily available). For example, based on previous

## Execution Time (s) vs. Number of Taxa

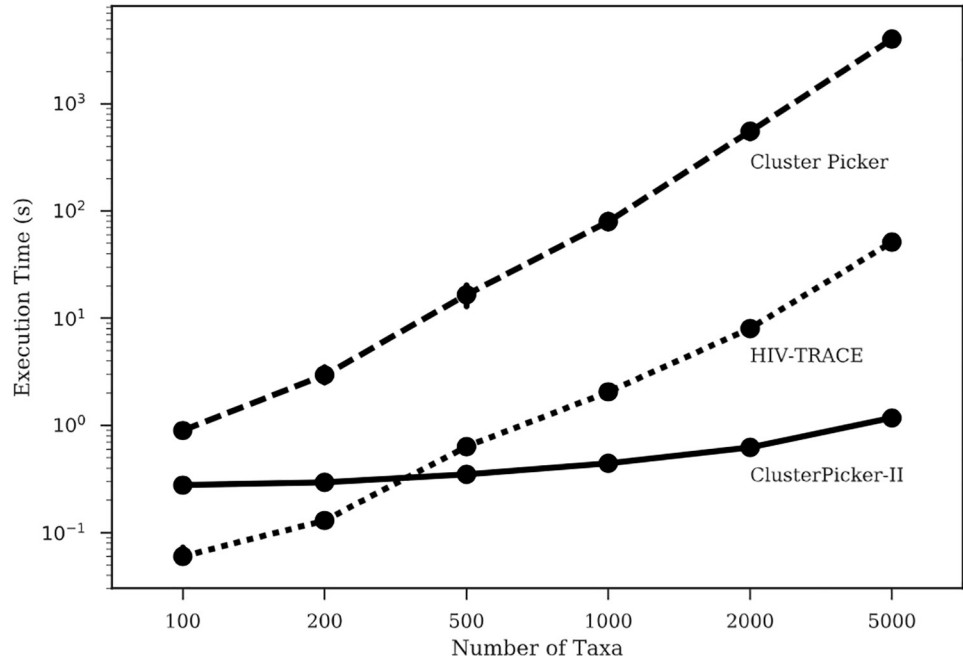

**Fig 6. Execution times of Cluster Picker, HIV-TRACE, and TreeCluster in log-scale.** Execution times (in seconds) are shown for each tool for various values of *n* sequences, with 10 replicates for each *n*. The full dataset was obtained by downloading all HIV-1 subtype B *pol* sequences (HXB2 coordinates 2,253 to 3,549) from the Los Alamos National Laboratory (LANL) database. All programs were run on a CentOS 5.8 machine with an Intel Xeon X7560 2.27 GHz CPU.

studies, MSA and tree inference on datasets with 10,000 sequences can take close to an hour using PASTA and 12 CPUs. Around a third of this time is spent on tree inference (e.g., see Fig 4 of [9]) and the rest is spent on the estimating alignment, which is also needed by most alternative clustering methods.

## Conclusion

We introduced TreeCluster, a method that can cluster sequences at the tips of a phylogenetic tree using several optimization objective functions. We showed that our linear-time algorithms can be used in several downstream applications, including OTU clustering, HIV transmission clustering, and divide-and-conquer alignment. Using the tree to build the clusters increases their internal consistency and improves downstream analyses.

## Supporting information

**S1 Fig. Comparison of various TreeCluster modes and Greengenes.** Clustering quality of Greengenes and various TreeCluster modes, where quality is measured as average pairwise distance within a cluster (the lower the better). The horizontal axis shows the number of clusters for a given method and a threshold value. TreeCluster OTUs based on Max-diameter and Sum-length options outperform Single-linkage option as well as Greengenes OTUs. Computation of Hamming distance based cluster diversity for $\alpha \geq 0.7$ did not complete within 24 hours and had to be terminated.
(PDF)

**S2 Fig. Tree distance versus Hamming distance.** On 16S data, the relationship between tree distances and Hamming distances cannot be established using the Jukes-Cantor formula (red curve).
(PDF)

**S3 Fig. An example showing that Mean-diameter min-cut partitioning is not conforming locality when $\alpha$ = 72, so it cannot be solved by a greedy algorithm analogous to Algorithm 1.** When a greedy algorithm is at the stage where it processes $u$, it makes the decision for cutting its children edges $(u, v)$ and $(u, a)$ based on the information available at the subtree rooted by $u$. When $\alpha$ = 72, (A) $T_1$ and (B) $T_2$ require different cut-sets ($\{(u, v)\}$ and $\{(u, a)\}$ respectively) for the optimal Mean-diameter partitioning despite the fact that the subtree rooted by $u$ remains unchanged in $T_1$ and $T_2$.
(TIFF)

**S1 Appendix. Proofs and supplementary algorithms.**
(PDF)

## Acknowledgments

This work was supported by the National Institutes of Health (NIH) subaward 5P30AI027767-28 to M.B., U.M, N.M, and S.M. and NSF grant NSF-1815485 and NSF-1845967 to M.B., U. M., and S.M. Computations were performed on the San Diego Supercomputer Center (SDSC) through XSEDE allocations, which is supported by the NSF grant ACI-1053575.

## Author Contributions

**Conceptualization:** Metin Balaban, Niema Moshiri, Siavash Mirarab.

**Data curation:** Metin Balaban, Niema Moshiri, Uyen Mai, Xingfan Jia.

**Formal analysis:** Metin Balaban, Uyen Mai, Siavash Mirarab.

**Funding acquisition:** Siavash Mirarab.

**Investigation:** Metin Balaban, Niema Moshiri, Uyen Mai, Xingfan Jia.

**Methodology:** Metin Balaban, Niema Moshiri, Siavash Mirarab.

**Project administration:** Siavash Mirarab.

**Resources:** Siavash Mirarab.

**Software:** Niema Moshiri.

**Supervision:** Siavash Mirarab.

**Validation:** Metin Balaban, Niema Moshiri, Uyen Mai, Xingfan Jia.

**Visualization:** Metin Balaban, Niema Moshiri, Uyen Mai, Siavash Mirarab.

**Writing – original draft:** Metin Balaban, Niema Moshiri, Siavash Mirarab.

**Writing – review & editing:** Metin Balaban, Niema Moshiri, Uyen Mai, Siavash Mirarab.

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
