## [Decision Letter · Decision Letter 0]

30 Jul 2019

TreeCluster: clustering biological sequences using phylogenetic trees

PONE-D-19-14871

Dear Dr. Balaban,

We are pleased to inform you that your manuscript has been judged scientifically suitable for publication and will be formally accepted for publication once it complies with all outstanding technical requirements.

With kind regards,

Serdar Bozdag, Ph.D.

Academic Editor

PLOS ONE

Reviewers' comments:

Reviewer's Responses to Questions

**Comments to the Author**

1. Is the manuscript technically sound, and do the data support the conclusions?

Reviewer #1: Yes

Reviewer #2: Yes

2. Has the statistical analysis been performed appropriately and rigorously? 

Reviewer #1: N/A

Reviewer #2: N/A

3. Have the authors made all data underlying the findings in their manuscript fully available?

Reviewer #1: Yes

Reviewer #2: Yes

4. Is the manuscript presented in an intelligible fashion and written in standard English?

Reviewer #1: Yes

Reviewer #2: Yes

5. Review Comments to the Author

Reviewer #1: This manuscript has been corrected since the original GLBIO submission, and still presents a well-reasoned and justified family of methods.

Reviewer #2: The authors considers three versions of clustering sequences based on phylogenies. As they rightly observe, similar problems have been considered in TCS context before, but somehow not used in bioinformatics.

It is good to see that all the problems have linear or near-linear time solutions, although the algorithms themselves use relatively standard algorithmic techniques. The authors also experimentally evaluate their approach in three contexts. I think the results are of sufficient interest for publication in this journal.

6. PLOS authors have the option to publish the peer review history of their article (what does this mean?). If published, this will include your full peer review and any attached files.

Reviewer #1: No

Reviewer #2: No

---

## [Editor Report · Acceptance letter]

9 Aug 2019

PONE-D-19-14918R1 

TreeCluster: clustering biological sequences using phylogenetic trees 

Dear Dr. Balaban:

I am pleased to inform you that your manuscript has been deemed suitable for publication in PLOS ONE. Congratulations! Your manuscript is now with our production department. 

With kind regards,

on behalf of

Dr. Serdar Bozdag 

Academic Editor

PLOS ONE